# Dynamic Executors of Bacterial Signals: Functional Versatility and Regulatory Networks of c-di-GMP Effectors

**DOI:** 10.3390/biom15101471

**Published:** 2025-10-17

**Authors:** Jia Jia, Ge Yun, Bingxin Liu, Xinxin Li, Meiling Jiang, Xinlu Yu, Jing Zhang, Yufei Han, Dan Liu, Junlong Zhao, Yuanyuan Wang, Gukui Chen

**Affiliations:** 1Key Laboratory of Resource Biology and Biotechnology in Western China, Ministry of Education, Faculty of Life Sciences and Medicine, Northwest University, Xi’an 710069, China; jiajia@nwu.edu.cn (J.J.); 202422217@stumail.nwu.edu.cn (G.Y.); 202422213@stumail.nwu.edu.cn (B.L.); 202324227@stumail.nwu.edu.cn (X.L.); 202332757@stumail.nwu.edu.cn (M.J.); 202332730@stumail.nwu.edu.cn (X.Y.); 202332705@stumail.nwu.edu.cn (J.Z.); 202422205@stumail.nwu.edu.cn (Y.H.); 202522342@stumail.nwu.edu.cn (D.L.); 20134631@nwu.edu.cn (J.Z.); 2College of Medical Technology, Shaanxi University of Chinese Medicine, Xianyang 712046, China

**Keywords:** the second messenger, c-di-GMP, diguanylate cyclase, phosphodiesterases, effectors, bacterial infections

## Abstract

Cyclic di-GMP (c-di-GMP), a universal second messenger in bacteria, orchestrates a wide array of essential life processes. Its intracellular dynamics are meticulously regulated by diguanylate cyclases (DGCs) and phosphodiesterases (PDEs), ensuring precise spatiotemporal control. The functional output of c-di-GMP signaling hinges on effector proteins—molecular decoders that translate c-di-GMP signals into specific cellular responses. This review systematically examines diverse classes of c-di-GMP effectors, using several representative bacterial species as model systems, to dissect their structural and mechanistic diversity. Particular emphasis is placed on their pivotal roles in bacterial pathogenicity, antibiotic tolerance, and host–pathogen interactions, offering fresh insights into the regulatory mechanisms underlying c-di-GMP signaling.

## 1. Introduction

Cyclic diguanosine monophosphate (c-di-GMP) is a widely distributed second messenger in bacteria. In 1987, researchers first defined c-di-GMP as an allosteric factor which can activate conformational change in cellulose synthase in *Komagataeibacter xylinus* [1,2]. So far, numerous studies have confirmed that c-di-GMP has multiple physiological regulatory functions in most bacteria, including regulating the cell cycle, biofilm formation, dispersion, bacterial motility, expression of virulence factors, etc., and enhances the community interaction of Gram-negative bacteria during biofilm growth by promoting the production of exopolysaccharides (EPS) and adhesins [3,4,5,6,7]. Furthermore, accumulating evidence indicates that c-di-GMP is pivotal in coordinating the switch between acute and persistent infection states, highlighting its role in bacterial pathogenesis dynamics [8].

Cyclic di-GMP (c-di-GMP) is synthesized by diguanylate cyclases (DGCs) containing a GGDEF domain, which catalyze the cyclization of two guanosine triphosphate (GTP) molecules. It is degraded into 5′-pGpG or two guanosine monophosphate (GMP) molecules by phosphodiesterases (PDEs) with EAL or HD-GYP domains [8,9,10]. DGCs, PDEs and their upstream regulatory factors coordinately regulate the dynamic changes in intracellular c-di-GMP concentration. Such concentration signals are recognized by various specific receptors or effector molecules, which directly or indirectly act on downstream targets to activate or repress signaling pathways. This process mediates multi-level regulation (transcriptional, translational, and post-translational) of bacterial physiological functions, enabling bacteria to adapt to fluctuating environmental conditions [2,11]. In summary, the activities of DGCs and PDEs determine the intracellular c-di-GMP levels, while effectors convert molecular concentration information into functional outputs or phenotypic changes. Therefore, c-di-GMP effectors serve as critical nodes in the complex signaling network, intricately regulating bacterial physiological functions.

Driven by breakthroughs in proteomics and X-ray crystallography, research on c-di-GMP effectors has deepened remarkably in recent years, enabling comprehensive functional and structural investigations. Reported effectors currently mainly include riboswitches and diverse proteins represented by transcription factors, PilZ domain proteins, and degenerate GGDEF/EAL domain proteins [12,13,14,15,16]. Researchers unexpectedly discovered that in *Escherichia coli*, a class of enzymatic EAL domain proteins (PdeR and PdeL) act as effectors to participate in c-di-GMP signal transduction [17]. A variety of c-di-GMP-binding proteins with high functional specificity have also been successively identified as effectors, including the *E. coli* EPS synthase and secretion proteins (Pga system), flagellar components and export ATPase (FliI) in plant-associated *Pseudomonas aeruginosa*, ribosome-modifying protein (RimK), and the key cell cycle regulator in *Caulobacter crescentus*—protein kinase/phosphatase (CckA) [18,19,20,21]. The ATPase MshE involved in the assembly of mannose-sensitive haemagglutinin (MSHA) pili in *Vibrio cholerae* and its homologous protein HxrA in *P. aeruginosa* also specifically bind to c-di-GMP [21,22,23,24]. Additionally, studies have demonstrated that c-di-GMP exhibits immunomodulatory effects in animals or humans, and several eukaryotic c-di-GMP effectors have been proven to be closely associated with multiple cellular physiological functions. For instance, stimulator of interferon genes (STING) can induce a type I interferon (IFN) response by binding to c-di-GMP, thereby functioning as an immune sensor. Moreover, the binding of STING to c-di-GMP can enhance the interaction between the C-terminal domain of STING and TANK-binding kinase 1 (TBK1), ultimately promoting the production of type I interferons [2,25,26]. Therefore, in this review, starting from the currently reported types of c-di-GMP effectors, we classify them into five categories based on differences in their structure or function. We then systematically and comprehensively summarize the structural and functional diversity of these effectors across various bacterial species, thereby providing theoretical support for the analysis of bacterial adaptive regulatory mechanisms and the development of novel antimicrobial therapies.

## 2. RNA-Based c-di-GMP Effectors: Riboswitches and Their Mechanisms

Riboswitches, also known as RNA molecular switches, represent an ancient gene regulatory mechanism that controls target gene expression by sensing changes in metabolite concentrations [27,28]. The c-di-GMP-binding riboswitch was the first genetic regulator discovered to recognize the second messenger [29]. To date, the structures and functions of two types of c-di-GMP-binding riboswitches have been validated: the first class, termed c-di-GMP-I, contains a bacterial RNA motif that regulates genes associated with environment, membrane, and motility; the second class (c-di-GMP-II) harbors an RNA motif encoding genes in bacteria and archaea that are widely involved in c-di-GMP synthesis, degradation, and signal transduction [29,30,31,32].

### 2.1. c-di-GMP-I Riboswitches: Structural Basis for Regulating Pathogenic Gene Expression in V. cholerae

In studies by Weinberg and Sudarsan, a highly conserved RNA domain termed GEMM (Genes for the Environment, Membranes and Motility) was discovered upstream of open reading frames (ORFs) encoding certain bacterial DGCs and PDEs, or genes potentially controlled by c-di-GMP. The high conservation and genomic distribution characteristics exhibited by GEMM RNA are typical features of c-di-GMP-I riboswitches [33,34]. GEMM RNAs adopt two major conformational types (Type I and Type II), distinguished by specific tetraloop and tetraloop receptor sequences (Figure 1a) [34]. Pathogenic *V. cholerae* harbors two Type I GEMM RNA sequences in its genome: one (*Vc1*) is located upstream of the *gbpA* gene, and the other (*Vc2*) is situated upstream of *VC1722*, a homologous gene of *tfoX* (a gene encoding a global transcription factor) (Figure 1b) [33,34,35]. Composed of 110 nucleotides, *Vc2* forms a binary saturated complex with c-di-GMP (*K_D_* = 1 nM, determined by Fluorescence Titration Method) exhibiting tighter binding affinity than the *E. coli* PilZ domain (*K_D_* = 840 nM, determined by Equilibrium Dialysis). When the intracellular concentration of c-di-GMP in bacteria increases, the *Vc2* riboswitch triggers higher expression of pathogenicity-related genes. Therefore, *Vc2* is referred to as a “gene-on switch” [34]. Notably, *gbpA* encodes a sugar-binding protein, which is critical for *V. cholerae* colonization of the mammalian intestine and subsequent cholera pathogenesis [36]. The *vieA* gene, encoding an EAL domain phosphodiesterase, is essential for bacterial infection. Sudarsan et al. used *vieA* to demonstrate that c-di-GMP-binding riboswitches respond to c-di-GMP level fluctuations, hypothesizing that the *Vc1* riboswitch senses VieA-mediated c-di-GMP depletion to promote the virulence-associated gene *gbpA* expression and initiate infection [34].

### 2.2. c-di-GMP-II Riboswitches: Controlling GTP-Mediated RNA Self-Splicing in Clostridioides difficile

Lee et al. identified an RNA motif through bioinformatics that is associated with c-di-GMP biosynthesis, degradation, and signal transduction genes. This motif was named c-di-GMP-II due to its structural features differentiating it from the previously discovered c-di-GMP-I riboswitch (Figure 2a) [32]. c-di-GMP-II predominantly exists in anaerobic Gram-positive bacteria of the genus *Clostridioides* [31]. Similarly to c-di-GMP-I, c-di-GMP-II typically localizes upstream of ORFs, consistent with its potential role in transcriptional termination-antitermination or translational control [31,32]. Using ^32^P-labeled c-di-GMP-II (84 nucleotides) from *Clostridioides difficile* strain 630, the study detected an apparent dissociation constant (*K_D_*) of ~200 pM for 1:1 c-di-GMP binding across a c-di-GMP concentration gradient [32,39,40]. Thus, c-di-GMP-II exhibits notably stronger binding affinity than c-di-GMP-I. We propose that this high affinity can minimize the probability of “regulatory errors”, ensuring that c-di-GMP signals are stably transmitted to downstream genes, thereby safeguarding the adaptability or pathogenicity of bacteria in complex environments. Additionally, this riboswitch lacks canonical riboswitch control structures and resides ~600 nucleotides upstream of its associated ORF (CD3246), creating a large gap for type I self-splicing ribozymes. The first splicing step in type I ribozymes involves a nucleophilic attack by the 3′-OH group of exogenous guanosine (e.g., from GTP) on the 5′-splice site, releasing the 5′ end of the intron decorated with extra guanosine (Figure 2b) [32,41].

Subsequently, to demonstrate that c-di-GMP controls the allosteric ribozyme function of c-di-GMP-II for self-splicing, Lee et al. used ^32^P-labeled riboswitches (864 nucleotides) containing the ribozyme domain and partial CD3246 ORF. In the absence of c-di-GMP, GTP-mediated cleavage occurred four nucleotides upstream of the start codon in the coding sequence, with no subsequent splicing. In contrast, c-di-GMP binding to c-di-GMP-II in the mRNA 5′ region induced secondary structure rearrangement, enabling GTP to attack a downstream site, ligate guanosine to the free 5′-end of the intron, and initiate the second-step attack on the 3′-splice site [32]. These findings indicate that c-di-GMP binding to c-di-GMP-II in vitro dictates the initial GTP attack site, thereby determining the occurrence of RNA self-splicing.

## 3. c-di-GMP-Responsive Transcription Factors: Structural Determinants and Regulatory Roles in Bacterial Physiology

Multiple studies have shown that a series of transcription factors in bacteria can bind to c-di-GMP, influencing biological processes such as bacterial adhesion, virulence, motility, and biofilm formation [11,13]. The enhancer-binding protein (EBP) transcription factor FleQ (PA1097) from the opportunistic pathogen *P. aeruginosa* is the first identified ATPase capable of binding to c-di-GMP [5,42,43]. The EBP transcription factor family serves as a critical regulator of bacterial functions, controlling bacterial motility, biofilm formation, quorum sensing, and virulence factor expression. Other members of this family are typically activated via phosphorylation, whereas FleQ activity is controlled by c-di-GMP [44]. FleQ contains a central AAA+ (ATPase Associated with diverse cellular Activities) ATPase-σ^54^ factor interaction domain flanked by a C-terminal helix-turn-helix (HTH) DNA-binding domain and a divergent N-terminal receptor domain. c-di-GMP interacts with the AAA+ ATPase domain of FleQ at a binding site distinct from the surface cavity-forming binding pockets. c-di-GMP binding inhibits FleQ ATPase activity, thereby altering its quaternary structure and transcriptional activity, leading to downregulation of FleQ-activated flagellar-related genes (e.g., *fleSR*) [42,45].

FleN (PA1454), another regulator of flagellar genes in *P. aeruginosa*, is an intracellular ATPase containing an atypical Walker A motif and lacks direct c-di-GMP binding activity [46,47]. Studies have revealed that the *fleN* gene is transcriptionally regulated by FleQ, while the presence of FleN enhances c-di-GMP-mediated inhibition of FleQ ATPase activity. The interplay between these two proteins coordinately regulates flagellar gene expression [42]. In the absence of c-di-GMP, FleQ directly represses the *pel* operon, which encodes machinery essential for EPS production (Figure 3a–c). As intracellular c-di-GMP levels rise, FleQ binds c-di-GMP and interacts with the accessory protein FleN, altering its binding configuration at the *pel* promoter to activate *pel* expression in a σ^70^-dependent manner [47] (Figure 3d). Recent structural elucidation of the FleQ-FleN complex in c-di-GMP-bound and -unbound states has deepened our understanding of nucleotide-dependent conformational switches and the fine-tuning of gene expression [48].

Studies have confirmed that FleQ homologs in other bacteria also exhibit c-di-GMP-responsive functions. For example, FlrA, the master regulator of flagellar biosynthesis in *V. cholerae*, is an EBP family transcription factor with ATPase activity. Srivastava et al. discovered that direct binding of c-di-GMP to FlrA impedes its interaction with the class II flagellar promoter *flrBC*, thereby disrupting the expression of flagellar biosynthesis regulators. Conversely, mutating both R135 and R176 of FlrA to histidine abolishes c-di-GMP binding, allowing FlrA to remain active even at high c-di-GMP concentrations [49]. In addition, *V. cholerae* harbors c-di-GMP receptor transcription factors lacking ATPase activity, such as the biofilm activator VpsT [50,51]. VpsT binds and is activated by high c-di-GMP, inducing expression of the *vpsI* and *vpsII* operons while repressing flagellar assembly genes. At high cell density, VpsT directly interacts with HapR to inhibit *hapR* transcription, thereby suppressing biofilm formation [50,52]. HapR serves as the master regulator of the quorum sensing (QS) cascade under high-density conditions [51]. AphA functions as both a virulence gene activator and the master regulator of low-density QS in *V. cholerae* and *Vibrio harveyi* [53]. VpsR, another NtrC-family transcription factor akin to FleQ, can bind c-di-GMP with a *K_D_* of 1.6 μM. VpsR binding to the *aphA* promoter region (−88 to −70) activates *aphA* transcription, whereas HapR binding at −85 to −57 region to repress it. Similarly, the *vpsT* promoter contains overlapping binding sites for VpsR (−136 to −118) and HapR (−144 to −123). Additionally, studies by Srivastava et al. have shown that VpsR is required for c-di-GMP activation of the *aphA* and *vpsT* promoters. Thus, they hypothesized that VpsR binding to c-di-GMP further promotes *aphA* and *vpsT* expression, which is crucial for c-di-GMP-mediated induction of the *aphA* and *vpsT* promoters [51].

In the genus *Mycobacterium*, two known c-di-GMP receptor regulators have been identified [54,55]. LtmA, the first characterized transcription factor with c-di-GMP signaling perception in *Mycobacteria*, acts as an activator that positively regulates the expression of redox gene clusters and hydrogen peroxide resistance, promoting bacterial growth under antibiotic stress conditions [54,56]. The other receptor transcription factor is HpoR, derived from *Mycobacterium smegmatis*, which exhibits an opposite redox negative regulatory function to LtmA [54,55]. HpoR can repress the expression of the *hpoR* operon, thereby increasing the sensitivity of *Mycobacteria* to hydrogen peroxide [57]. Studies have found that these two transcription factors exhibit high amino acid sequence homology (43%) and have direct physical interactions. High levels of c-di-GMP in bacteria stimulate the positive regulation of LtmA, promote interactions between the two regulators, further enhance the DNA-binding ability of LtmA, and reduce the inhibitory activity of HpoR [56]. Thus, when bacteria are exposed to oxidative stress conditions, c-di-GMP triggers the antioxidant defense of *Mycobacteria* by coordinating transcription factors with distinct functions. Subsequent studies by Hu et al. have shown that DevR, the transcription factor in the DevR-DevS two-component system of the bacterial oxidative stress response, is a novel c-di-GMP receptor. They found that DevR can sense c-di-GMP signals through its C-terminal domain. c-di-GMP does not directly affect the DNA-binding activity of DevR; however, it indirectly activates DevR’s DNA-binding affinity by stimulating DevR phosphorylation via the DevS kinase. High c-di-GMP levels induce the expression of the *devR* operon in *Mycobacterium smegmatis* and enhance the survival of mycobacteria under oxidative stress. Deletion of either DevR or its two-component kinase DevS significantly impairs the stimulatory effect of c-di-GMP on the oxidative stress tolerance of *Mycobacteria* [58].

In the phytopathogen *Xanthomonas campestris*, the expression of a repertoire of virulence genes is regulated by both c-di-GMP and XcCLP, a catabolite activation protein (CAP)-like transcription factor [59]. XcCLP, featuring an N-terminal β-barrel domain and a C-terminal DNA-binding domain, functions as a global regulator within the CRP (cAMP receptor protein)/FNR (fumarate nitrate reductase regulator) superfamily [59,60]. Studies have demonstrated that XcCLP directly binds c-di-GMP with a *K_D_* of 3.5 μM, inducing a conformational change that disrupts the expression of downstream virulence genes [59].

## 4. The PilZ Domain Fold: c-di-GMP Binding and Allosteric Regulation

PilZ domain proteins represent the most widely distributed class of c-di-GMP effectors known to date, conserved across bacterial species and involved in regulating diverse physiological functions including bacterial pathogenicity, motility, EPS production, and biofilm formation [11].

In bacterial biofilms, cellulose—a common EPS—is synthesized by the cellulose synthase BcsA-BcsB complex and translocated across the inner membrane [11]. In studies on cellulose biosynthesis in *K. xylinus*, Amikam and Galperin discovered that c-di-GMP acts as an allosteric activator of cellulose synthase by binding to the BcsA subunit, thereby stimulating cellulose production [61]. BcsA from *Rhodobacter sphaeroides*, homologous to eukaryotic cellulose synthases, contains eight transmembrane (TM) helices and an intracellular glycosyltransferase (GT) domain between TM helices 4 and 5 (Figure 4a,b) [62,63]. Among them, a conserved c-di-GMP recognition and binding amino acid sequence, namely the PilZ domain, exists in the C-terminal helical region (TM8) (Figure 4a,b). This region contains an “RxxxR” motif located in the flexible linker, followed by a β-sheet or β-barrel structure with a “DxSxxG” motif [62]. The β-barrel leans against the intracellular GT domain and is connected to the C-terminal TM8 through the TM8-β-barrel linker [61]. The TM8-β-barrel linker also interacts with the “gating loop” of BcsA, which passes through the opening of the GT domain and faces the cytoplasm, thereby preventing access to the catalytic pocket of the enzyme in the unstimulated or “resting” state [62]. Morgan et al. elucidated the mechanism by which c-di-GMP activates bacterial cellulose synthases through structural characterization of the BcsA-BcsB complex binding c-di-GMP during cellulose biosynthesis and translocation in *R. sphaeroides*. The crystal structure of c-di-GMP-activated BcsA-BcsB reveals that c-di-GMP releases the enzyme from autoinhibition by breaking salt bridges, generating a constitutively active cellulose synthase. These salt bridges typically link the conserved gating loop of BcsA, which controls substrate access to the active site [64].

To the present, researchers have identified multiple c-di-GMP effectors containing PilZ domains in *P. aeruginosa*: Alg44, PilZ, PA2560, FlgZ, MapZ, HapZ, PA0012, PA4324, PA2989, and FimW [65,66,67,68,69,70,71,72]. The gene encoding Alg44 is located within a conserved gene cluster responsible for synthesizing the EPS alginate [73]. Alginate plays a critical role in *P. aeruginosa* chronic infections in cystic fibrosis patients [74]. Alg44 comprises an N-terminal PilZ domain and a C-terminal membrane fusion domain [75]. After Merighi et al. mutated several conserved amino acids (R17, R21, D44, S46) in its N-terminal PilZ domain to alanine, they used the filter binding assay to determine the binding ability of the Alg44^mutant^ (Alg44^R17A^, Alg44^R17A,R21A^, Alg44^D44A^, Alg44^R21A^) to c-di-GMP in vitro, and also detected the alginate phenotype of these mutants. It was found that the Alg44^mutant^ lost their c-di-GMP binding ability and failed to produce alginate [65]. Furthermore, in experiments testing the c-di-GMP binding capacity of PilZ domain proteins from *P. aeruginosa* in vitro, Merighi et al. found that only one protein lost c-di-GMP binding activity—PilZ itself—which lacks critical conserved residues in the “DxSxxG” motif essential for c-di-GMP binding [61,65]. PilZ is involved in regulating Type IV pili (T4P) assembly, serves as a key protein for T4P-mediated bacterial twitching motility, and participates in EPS synthesis and secretion [76,77]. Hendrix et al. discovered that during screening for host factors affecting phage infection, PA2560 (PlzR, PilZ regulator) regulates the ATPase PilB by directly binding to the T4P chaperone PilZ, leading to impaired T4P assembly. Moreover, the *plzR* promoter is induced by c-di-GMP [78]. Another PilZ domain protein involved in T4P assembly, FimW, is activated by c-di-GMP, thereby triggering bacterial surface colonization and virulence induction [72].The rotation of the polar flagellum in *P. aeruginosa* is controlled by two distinct motor complexes, MotAB and MotCD. MotAB inhibits bacterial swarming, whereas MotCD promotes it [79,80]. Baker et al. demonstrated that FlgZ, a c-di-GMP-binding protein, suppresses swarming by interacting with the flagellar regulator MotC. Both c-di-GMP binding and the conserved PilZ domain are essential for FlgZ-mediated swarming inhibition and the FlgZ-MotC interaction [65]. Bense et al. further confirmed that FlgZ-mediated motility downregulation is finely tuned through three independent mechanisms: First, *flgZ* is transcribed independently of *flgMN* during the stationary phase to increase intracellular FlgZ levels; second, c-di-GMP binding promotes polar localization of FlgZ; third, the polar anchor protein FimV contributes to FlgZ anchoring [81]. Xu et al. from South China Agricultural University has explicated the functional mechanisms of MapZ and HapZ in *P. aeruginosa*: MapZ directly interacts with the chemotaxis methyltransferase CheR1 to regulate the switching of flagellar motors; HapZ acts in concert with the histidine kinase SagS to regulate the two-component signaling system [69,70,82,83]. Additionally, the team screened the phenotypes of *PA0012* and *PA4324* mutants, showing that mutations in both genes enhance swarming motility of *P. aeruginosa* while weakening its ability to infect lettuce [84]. Based on this, they further illustrated the mechanism by which PA0012 (TssZ, a Type VI secretion system-associated PilZ protein) regulates the Type VI secretion systems (T6SS) in *P. aeruginosa*: As a suppressor, TssZ interacts with the transcriptional regulator HinK to affect bacterial swarming motility and T6SS-mediated killing activity in a QS regulator PqsR-dependent manner. The TssZ-HinK interaction is independent of c-di-GMP but influenced by its concentration, with low c-di-GMP levels promoting their interaction [68]. Subsequently, their study on the pathogenic-related biological functions of PA2989 revealed that mutation of *PA2989* affects bacterial swarming motility and EPS, and its deletion may impact intracellular c-di-GMP concentrations. However, the mechanism by which PA2989 regulates the pathogenicity of *P. aeruginosa* through c-di-GMP remains unknown [85].

The PilZ domain protein YcgR in *E. coli* and its homologs in *Salmonella enterica*, *Bacillus subtilis*, and *C. crescentus* have been shown to bind c-di-GMP and act as inhibitors of cellular motility [86,87,88,89]. Studies have shown that these PilZ domain proteins impair flagellar function by interacting with flagellar motors and switch complexes. In *E. coli*, YcgR interacts with three distinct flagellar regulatory proteins: MotA, FliG, and FliM [86,87]. Among them, the YcgR-MotA interaction requires c-di-GMP binding, while interactions between YcgR and FliG/FliM are most pronounced in the presence of c-di-GMP. In *B. subtilis*, the YcgR homolog MotI (previously known as YpfA or DgrA) can interact with MotA [89,90]. The motility control function of DgrA in *C. crescentus* is associated with the level of class II flagellar protein FliL [91].

The YajQ family protein XC_3703 from *X. campestris* is a potential c-di-GMP receptor with high binding affinity (*K_D_* = 2 μM). Mutation of XC_3703 impairs plant virulence and biofilm formation in *X. campestris* XC_3703 interacts with the LysR-type transcription factor XC_2801 to enhance the binding ability of XC_2801 to target virulence gene promoters. However, this effect can be reversed by c-di-GMP, where binding of c-di-GMP to XC_3703 inhibits the interaction between the XC_3703/XC_2801 complex and DNA, thereby blocking the transcription of class II flagellar gene *flhBA*. Genetic and functional analyses of YajQ family members in *P. aeruginosa* and *Stenotrophomonas maltophilia* further revealed that these proteins not only bind c-di-GMP specifically but also enhance bacterial pathogenicity [92,93]. Building on these findings, Han et al. discovered a similar YajQ-LysR system in the crop-protective bacterium *Lysobacter enzymogenes* OH11, where this system inhibits transcription of the *heat-stable antifungal factor* (*HSAF*) operon—a locus encoding an antifungal antibiotic—by increasing c-di-GMP concentrations. The system includes a c-di-GMP receptor protein CdgL, a homolog of *X. campestris* YajQ, which similarly binds to a LysR-type transcription factor essential for activating *HSAF* operon transcription [93,94].

## 5. GGDEF/EAL Domain Proteins: Degenerate Domains and Triggered Phosphodiesterases

### 5.1. Structural Basis of Degenerate Domain Proteins: Conserved c-di-GMP Binding Motifs Distinct from Catalytic Sites

c-di-GMP controls a series of important cellular functions in both eukaryotes and prokaryotes. Studies have found that the cell cycle of *C. crescentus* is regulated by c-di-GMP and its effectors. During the G1 to S transition in the interphase of cell mitosis, c-di-GMP promotes the progression of the bacterial cell cycle by mediating the specific degradation of the transcription factor CtrA that inhibits replication initiation [5,16]. During this process, CtrA relies on a c-di-GMP-binding protein PopA localized at the cell pole to be directed to this subcellular site, followed by rapid degradation by its corresponding protease ClpXP [16,95,96]. PopA (paralog of pleD), a response regulator with a GGDEF domain encoded by ORF CC1842 in *C. crescentus*, harbors a conserved high-affinity c-di-GMP binding motif (RxxD) at its I-site within the GGDEF domain. However, the degenerate motif at the catalytic A-site renders it incapable of synthesizing c-di-GMP [16]. The RxxD motif in diguanylate cyclases WspR and PleD also serves as the primary c-di-GMP binding site, located distal to the active site loop of the GGDEF domain. Through interaction with c-di-GMP via their RxxD motifs, WspR and PleD achieve product inhibition of DGCs, thereby regulating c-di-GMP signaling and cellular homeostasis [97,98].

FimX is a large protein containing four domains: REC, PAS, GGDEF, and EAL. Three of these domains are functionally degenerate: The REC receiver domain lacks the phosphorylation site of its cognate histidine kinase, the GGDEF domain lacks DGC activity, and the EAL domain retains c-di-GMP binding ability despite losing PDE activity due to the conserved ExLxR motif [99,100]. Studies have shown that in *Pseudomonas* and *Xanthomonas*, FimX acts as a c-di-GMP effector to regulate bacterial vital activities [101,102,103]. In *P. aeruginosa* and *Xanthomonas citri*, c-di-GMP-bound FimX interacts with the ATPase PilB to enhance its ATPase activity, thereby positively regulating T4P assembly and twitching motility. Further studies have revealed that FimX, PilZ, and ATPase PilB in *X. citri* forms a stable ternary complex, which is involved in the regulation of T4P biosynthesis [100,101].

There is a c-di-GMP-regulated signaling system in the environmental bacterium *Pseudomonas fluorescens*, which can regulate biofilm formation according to the level of inorganic phosphate in the environment. The system includes the transmembrane c-di-GMP effector LapD and the periplasmic protease LapG. LapD utilizes its degenerate EAL domain (ExLxR motif) to bind c-di-GMP and regulates LapG to control the protease activity toward the cell-surface adhesin LapA [102]. CdrA, a protein involved in intercellular aggregation and biofilm maturation in *P. aeruginosa*, as a substrate of LapG, and is cleaved from the cell surface by LapG, thereby playing a role in biofilm formation and dispersal. The LapDG system co-regulates CdrA localization in response to intracellular c-di-GMP concentration changes: CdrA remains in the whole-cell protein fraction when LapG is absent, but appears in the culture supernatant when LapG is present [103,104,105].

### 5.2. Triggered Phosphodiesterases as Transcriptional Switches: Targeting Factor Binding for Gene Regulation

In *E. coli*, the c-di-GMP-specific phosphodiesterase PdeR is considered as the first identified “triggered enzyme” involved in c-di-GMP signaling, thus named a “triggered phosphodiesterase”, which acts as a novel c-di-GMP effector. While PdeR shares the c-di-GMP allosteric mechanism with other effectors, its distinguishing feature lies in the concurrent execution of ligand binding and catalytic degradation—a dual functionality that enables triggered PDEs to not only sense c-di-GMP but also act as self-regulating modules, dynamically adjusting c-di-GMP levels through an intrinsic negative feedback loop [17]. CsgD serves as an indirect regulator of cellulose synthase activity. MlrA, a highly specific activator of csgD transcription initiation, relies on the diguanylate cyclase DgcM (formerly YdaM) for its activity. As a key regulatory factor controlling the molecular switch for biofilm synthesis in *E. coli*, PdeR exerts dual functions: At low c-di-GMP levels, PdeR inhibits both DgcM and the transcription factor MlrA through direct interactions, thereby preventing the expression of the biofilm regulator CsgD. When c-di-GMP levels rise, PdeR binds and degrades c-di-GMP, releasing DgcM and MlrA. This allows DgcM to act as a direct co-activator of MlrA, driving the transcriptional activation of the *csgDEFG* operon while simultaneously producing c-di-GMP [106,107].

PdeL is another validated triggered phosphodiesterase in *E. coli*, which contains an N-terminal LuxR-like domain with a HTH motif linked to an EAL domain. Binding of c-di-GMP promotes dimerization of the purified EAL domain, and the dimer interface facilitates the formation of an active catalytic center [108,109]. At high c-di-GMP concentrations, the binding and degradation of c-di-GMP interfere with PdeL-mediated transcriptional activation of pdeL. Conversely, PdeL can directly bind to the *pdeL* promoter region to activate its own expression [110,111]. In this regulatory paradigm, the inhibitory effect of c-di-GMP on the expression of the degradation enzyme PdeL and the positive autoregulation of PdeL itself constitute a dual positive feedback loop. This combinatorial mechanism enables the rapid and efficient switching of cellular c-di-GMP levels to a low-concentration state, thereby promptly terminating the expression of biofilm-related functions [17].

## 6. The Role of c-di-GMP Effectors with High Functional Specificity in Bacterial Life Activities

c-di-GMP regulates a variety of cellular activities, including the production of EPS, pilus biogenesis, flagella-based motility, and RNA degradation in *E.coli*. Through allosteric regulation, c-di-GMP activates the synthesis of poly-β-1,6-N-acetylglucosamine (poly-GlcNAc), a major component of the extracellular matrix in biofilms [18,112]. Poly-GlcNAc is synthesized and secreted by the transmembrane Pga system encoded by the *pgaABCD* operon. Among these members, PgaA and PgaB are essential for poly-GlcNAc export, while PgaC and PgaD are responsible for its synthesis [18,113,114,115]. Steiner et al. revealed the mechanism by which c-di-GMP regulates EPS production: c-di-GMP directly binds to two inner membrane components of the Pga system, PgaC and PgaD, stabilizing their interaction and consequently stimulating glycosyltransferase (GT) activity. At low c-di-GMP concentrations, PgaD exhibits instability, fails to bind to PgaC and is rapidly degraded. Consequently, when cellular c-di-GMP levels decline, the rapid turnover of PgaD leads to irreversible inactivation of the Pga system (which is responsible for the synthesis and secretion of poly-GlcNAc), temporarily decoupling the synthesis and secretion of poly-GlcNAc from the regulation of c-di-GMP levels. At this point, the cell needs to activate the derepressed carbon storage regulator (Csr) pathway to support the resynthesis of all Pga components, thereby restoring the synthesis of poly-N-acetylglucosamine (poly-GlcNAc). This mechanism ensures that bacteria can still maintain normal life activities when short-term fluctuations occur in c-di-GMP levels [18].

*P. fluorescens* is a widely distributed soil bacterium that forms symbiotic relationships with plant species, which can colonize the rhizosphere and phyllosphere of various plants non-specifically, promote plant growth, and exhibit strong antifungal activity [116,117]. *Pseudomonas syringae* is a plant pathogenic bacterium responsible for a variety of important plant diseases, whose secretions disrupt plant defense mechanisms and invade host plants through open stomata and wounds on the plant surface [118,119]. Flagella and type III secretion systems (T3SS) are the two most critical organelles enabling effective colonization of hosts by symbiotic or pathogenic *Pseudomonas* species [120]. The assembly of bacterial flagella is tightly regulated. The flagellar export ATPase FliI is essential for the formation of functional flagella; it forms the flagellar export apparatus together with FliH and FliJ, and plays a role in protein export [121,122,123]. To investigate the role of c-di-GMP in plant-associated *P. aeruginosa*, Trampari et al. performed affinity capture screening of c-di-GMP-binding proteins in the model bacterium *P. fluorescens* SBW25. The results revealed that FliI is an effector that specifically binds to c-di-GMP. Subsequently, the binding sites between the two were further predicted using a combination of mass spectrometric analysis and in silico simulations: c-di-GMP likely binds to a pocket of highly conserved residues at the interface of the two FliI subunits. The binding of FliI to c-di-GMP is not restricted to *P. fluorescens*; it also occurs in FliI homologs from other plant-associated *P. aeruginosa* species, such as the T3SS export ATPase HrcN and the T6SS export ATPase ClpB2 in *P. syringae* [11,19]. In addition, studies have elucidated the function of the RimABK system: the ribosome-modifying protein RimK tightly regulates its own activity through binding to the small regulatory proteins RimA and RimB, as well as to c-di-GMP. This novel regulatory mechanism dynamically influences interactions between bacteria and their hosts. Deletion of the *rimK* gene impairs motility, virulence, and the ability to colonize and infect plants in various *Pseudomonas* species. Changes in intracellular RimK activity enable *Pseudomonas* to respond to environmental stress by altering ribosome properties, thereby triggering adaptive phenotypic responses to the surrounding environment [21].

The previous text briefly described the function of the cell cycle transcriptional regulator CtrA in cellular replication: CtrA is phosphorylated and activated during the G1 phase of cell division, thereby binding to the replication origin (*Cori*) and preventing the initiation of replication [124]. During differentiation into stalked cells, CtrA is inactivated and replication is initiated. The activity of CtrA is controlled by the bifunctional cell cycle histidine kinase CckA, which phosphorylates and activates CtrA via the phosphotransferase ChpT. CckA exhibits kinase activity during the G1 phase but switches to significant phosphatase activity during the G1 to S transition, thereby reversing the phosphate flow in the CckA-ChpT-CtrA cascade and inactivating CtrA. The phosphatase or kinase activity of CckA is tightly regulated by c-di-GMP: when c-di-GMP binds to the catalytic and ATP-binding domains of CckA, kinase activity is inhibited while phosphatase activity is activated. By controlling this key cell cycle kinase, c-di-GMP promotes the G1 to S transition in *C. crescentus*, functioning analogously to cyclins in eukaryotes, which drive the cell cycle by regulating the activity of cyclin-dependent kinases [125]. Moreover, the spatial regulation of CckA activity by c-di-GMP also promotes asymmetric replication in prospective daughter cells. In pre-divisional cells, CckA localizes to the opposing poles of the cell, exerting kinase and phosphatase activities at the flagellar pole and stalked pole, respectively. This leads to the formation of an intracellular concentration gradient of phosphorylated CtrA and triggers the asymmetric initiation of replication [126,127,128].

## 7. Conclusions and Outlook

As the core executors of the bacterial second messenger c-di-GMP signaling pathway, different types of effectors—such as RNA riboswitches, transcription factors, PilZ domain-containing proteins, GGDEF or EAL domain-containing proteins, and other functionally specific proteins—utilize diverse mechanisms including RNA regulation, protein–protein interaction, allosteric regulation, and product inhibition to precisely control key life activities such as biofilm formation, motility, virulence gene expression, cell division processes, and host parasitism (Figure 5). Through comprehensive elaboration on the functional mechanisms of these various c-di-GMP effectors, three key aspects have been systematically revealed, as laid out below, establishing the role of effectors as central schedulers in bacterial environmental adaptation mechanisms.
(1)The evolution and divergence of effector domains lay the molecular foundation for their functional diversity. For example, c-di-GMP acts as an allosteric activator of cellulose synthase by binding to the conserved active site in the PilZ domain, subsequently stimulating cellulose synthesis. Effectors containing degenerate GGDEF or EAL domains, although losing their intrinsic enzymatic activity as DGCs or PDEs, can still bind to c-di-GMP due to their specific c-di-GMP-binding conserved motifs. Trigger PDEs with normally catalytically active EAL domains regulate the expression of biofilm-related genes through allosteric regulation and product inhibition.(2)A single effector can integrate multiple signaling pathways (Figure 3). For instance, PilZ is involved in two distinct signaling pathways: one regulating EPS synthesis and another integrating T3SS and flagellar functions. The transcription factor FleQ, on the one hand, undergoes changes in its quaternary structure and transcriptional activity upon binding to c-di-GMP, thereby repressing the expression of flagellar-related genes; on the other hand, FleQ cooperates with FleN to regulate EPS production. Additionally, another transcription factor, VpsT, inhibits flagellar assembly under high c-di-GMP levels and suppresses biofilm formation under high cell density conditions.(3)Functional redundancy among effectors may constitute a bacterial barrier against antibiotic resistance (Table 1). For example, the integration and activity regulation of bacterial motility-related flagella involve multiple effectors with distinct structures, such as FlgZ, YcgR, and FliI. Transcription factors such as FleQ, GGDEF domain-containing proteins such as PelD, PilZ domain-containing proteins such as BcsA, Alg44, and PA2989, as well as functionally specific proteins such as PgaCD, synergistically regulate the synthesis and secretion of EPS through distinct c-di-GMP regulatory networks.

Although the structures and functions of these effectors have been fully characterized, there remain some unresolved molecular mechanisms. For instance, PA2989 influences bacterial swarming motility and EPS production in *P. aeruginosa*, but the pathogenic mechanism by which PA2989 regulates bacterial virulence via c-di-GMP remains unknown [85]. While studies have found that the trigger phosphodiesterase PdeL is autostimulated and further accelerates the degradation of c-di-GMP, whether this mechanism has physiological significance and its triggering timing remain to be confirmed by subsequent studies [17]. Moreover, whether PdeL regulates other target genes beyond its own gene remains to be investigated. Early studies primarily documented the post-translational regulatory roles of c-di-GMP. However, recent characterization of DNA-binding proteins such as FleQ/FlrA, VpsT, VpsR, LtmA, CLP, BldD, BrlR, and MrkH has provided clear evidence of c-di-GMP’s involvement in transcriptional regulation [42,43,44,45,46,47,48,49,50,51,54,55,57,59,129,130,131,132]. Notably, while most of these transcriptional regulators bind DNA through canonical HTH domains, the MrkH protein combines its c-di-GMP-binding PilZ domain with a novel DNA-binding domain [131]. Proteins with diverse domain architectures that link PilZ domains to uncharacterized or poorly defined domains clearly represent potential research directions for novel c-di-GMP receptors, offering highly attractive candidates for future experimental investigations.

The ability of c-di-GMP effectors to integrate multiple pathways as signaling hubs represents an efficient adaptive strategy evolved by bacteria, with their molecular mechanisms relying on domain plasticity, transcriptional regulation, post-translational modifications, and multi-ligand interactions. In-depth analysis of such integration modes not only reveals the core logic of bacterial pathogenicity but also provides targets for developing “multi-pathway interference” antimicrobial strategies. For example, designing bifunctional inhibitors that simultaneously block the binding of effectors to both c-di-GMP and host signaling molecules is expected to overcome the limitations of drug resistance associated with single targets. In future studies, live-cell dynamic imaging technology can be specifically integrated to achieve full-process tracking of the spatiotemporal localization and dynamic interactions of effectors within bacterial hosts to the greatest extent possible. Computational biology approaches should be employed to construct interaction models between effectors and host target proteins, thereby deciphering the regulatory mechanisms mediated by allosteric effects. Additionally, synthetic biology tools can be utilized to validate the functions of effectors by reconstructing or interfering with their action pathways. Based on the combined application of the aforementioned technologies, it will be feasible to systematically decode the multi-dimensional regulatory networks of effectors at the levels of “protein interaction—signal pathway regulation—bacterial pathogenic phenotype”. Ultimately, this will provide theoretical and technical support for the development of new paradigms for the precise intervention of bacterial infections, such as “small-molecule inhibitors targeting key interaction sites of effectors” and “gene therapy strategies blocking effector regulatory pathways”.

**Table 1 biomolecules-15-01471-t001:** Different types of bacterial c-di-GMP effectors.

Effectors	Types	Sources	c-di-GMP Binding Domain	Binding Affinity (*K_D_*)	Functional Effects of c-di-GMP Binding	References
c-di-GMP-I	RNA Riboswitches	*V. cholerae*	Type I GEMM RNA (*Vc2*)	1 nM	Activate the expression of genes associated with intestinal colonization	[33,34,35,36]
c-di-GMP-II	RNA Riboswitches	*C. difficile*	mRNA 5′-region	200 pM	Determine the site of RNA self-splicing	[31,32,39,40,41]
FleQ	Transcription Factors	*P. aeruginosa*	Receptor domain(REC) at N-terminus	7 μM	Repress flagellar -related genes expression, activate EPS production	[5,42,43,44,45]
BrlR	Transcription Factors	*P. aeruginosa*	Gyrl-like Domain at C-terminus	2.2 μM	Enhance the high-level drug resistance of biofilms	[132]
FlrA	Transcription Factors	*V. cholerae*	REC at N-terminus	0.38 μM	Interfere with flagellar biosynthesis	[49]
VpsT	Transcription Factors	*V. cholerae*	REC, HTH	3.2 μM	Promote biofilm synthesis and repress flagellar assembly genes expression	[50,51,52]
VpsR	Transcription Factors	*V. cholerae*	predicted ATP binding domain	1.6 μM	Activate biofilm synthesis	[51]
LtmA	Transcription Factors	*M. smegmatis*	TetR-type HTH domain	0.83 μM	Positively regulate the expression of redox gene clusters, enhance bacterial resistance to hydrogen peroxide	[54,55,56,57]
HpoR	Transcription Factors	*M. smegmatis*	---	1.78 μM	Negatively regulate the expression of redox gene clusters, increase bacterial sensitivity to hydrogen peroxide	[56,57]
DevR	Transcription Factors	*M. smegmatis*	C-terminus	1.96 μM	Increase survival rate under oxidative stress	[58]
XcCLP	Transcription Factors	*X. campestris*	β-barrel domain at N-terminus	3.5 μM	Interfere with downstream virulence genes expression	[59,60]
BldD	Transcription Factors	*Streptomyces coelicolor*	---	2.5 μM	Inhibit sporulation genes expression	[129]
MrkH	Transcription Factors	*Klebsiella pneumoniae*	PilZ domain at N-terminus	0.107 μM	Promote biofilm formation	[131]
FsnR	Transcription Factors	*Stenotrophomonas maltophilia*	REC, HTH	3.43 μM	Induce flagellar genes expression	[133]
CdbA	Transcription Factors	*Myxococcus xanthus*	RHH DNA binding domain	~0.083 μM	Promote chromosome formation	[134]
BcsA	PilZ domain	*K. xylinus*	PilZ	0.98 μM	Regulate cellulose synthesis	[61,62,63,64]
Alg44	PilZ domain	*P. aeruginosa*	PilZ	---	Regulate alginate synthesis	[69,70]
FimW	PilZ domain	*P. aeruginosa*	PilZ	---	Regulate T4P assembly	[72]
PilZ	PilZ domain	*P. aeruginosa*	PilZ	---	Regulate T4P assembly, mediate bacterial twitching motility, regulate EPS synthesis	[76,77]
PlzR	PilZ domain	*P. aeruginosa*	PilZ	---	Regulate T4P assembly	[78]
FlgZ (PA2560)	PilZ domain	*P. aeruginosa*	PilZ	---	Inhibits bacterial swarming	[81]
MapZ	PilZ domain	*P. aeruginosa*	PilZ	---	Regulate the switching of flagellar motors	[82]
HapZ	PilZ domain	*P. aeruginosa*	PilZ	2.0 μM	Regulate two-component signal transduction	[83]
TssZ (PA0012)	PilZ domain	*P. aeruginosa*	PilZ	---	Affect bacterial swarming motility and T6SS-mediated bacterial killing activity	[68]
PA4324	PilZ domain	*P. aeruginosa*	PilZ		Affect bacterial swarming motility and virulence	[84]
PA2989	PilZ domain	*P. aeruginosa*	PilZ		Affect bacterial swarming motility and EPS	[85]
YcgR	PilZ domain	*E.coli*	PilZ	---	Impair flagellar function and inhibit bacterial motility	[86,87]
MotI	PilZ domain	*B. subtilis*	PilZ	---	Impair flagellar function and inhibit bacterial motility	[89,90]
DgrA	PilZ domain	*C.crescentus*	PilZ	---	Impair flagellar function and inhibit bacterial motility	[91]
YajQ (XC_3703)	PilZ domain	*X. campestris*	PilZ	2.0 μM	Activate virulence genes expression, enhance bacterial pathogenicity	[92,93]
CdgL	PilZ domain	*L. enzymogenes*	PilZ	---	Activate HSAF expression	[94]
Tlp1	PilZ domain	*Azospirillum brasilense*	PilZ	---	promote the bacterial continuous motility	[135]
Cbp1	PilZ domain	*Azorhizobium caulinodans*	PilZ	14.94 μM	Regulate bacterial motility, biofilm, virulence	[136]
CdbS	PilZ domain	*M. xanthus*	PilZ	~1.4 μM	Interfere with bacterial chromosome organization and accelerate cell death under heat stress	[137]
PlzC/PlzD	PilZ domain	*V. cholerae*	PilZ	0.1–0.3 μM	Regulate bacterial motility, biofilm, virulence	[138,139]
PopA	GGDEF/EAL domain	*C.crescentus*	GGDEF	2 μM	Regulate cell cycle	[16,95,96]
PleD	GGDEF/EAL domain	*C.crescentus*	GGDEF	---	---	[97,98]
WspR	GGDEF/EAL domain	*C. crescentus,* *P. aeruginosa*	GGDEF	---	Regulate biofilm formation	[97,98]
PelD	GGDEF/EAL domain	*P. aeruginosa*	GGDEF	0.5–1.9 μM	Produced Pel polysaccharide	[140]
FimX	GGDEF/EAL domain	*P. aeruginosa,* *Xanthomonas citri*	EAL	0.1–0.2 μM	Regulate bacterial twitching motility, T4P synthesis, biofilm formation, virulence genes expression	[99,100,101]
LapD	GGDEF/EAL domain	*Pseudomonas fluorescens*	EAL	1.9 μM	Control cell adhesion and biofilm formation	[102,103,104,105]
PdeR	GGDEF/EAL domain	*E.coli*	EAL	---	Regulate biofilm formation	[106,107]
PdeL	GGDEF/EAL domain	*E.coli*	EAL	---	Regulate biofilm formation	[110,111]
PgaC/PgaD	Function- specific effector	*E.coli*	---	---	Induce biofilm formation	[113,114,115]
CobB_L_	Function- specific effector	*E.coli*	N-terminus	4.7 μM	Control energy metabolism, chemotaxis, and DNA supercoiling	[141,142]
FliI	Function- specific effector	*P. fluorescens*	---	---	Control flagellar export	[121,122,123]
HrcN	Function- specific effector	*Pseudomonas syringae*	---	---	Control flagellar export	[11,19]
ClpB2	Function- specific effector	*P. syringae*	---	---	Control flagellar export	[11,19]
RimK	Function- specific effector	*P. fluorescens*, *P. syringae*	---	---	Affect bacterial motility, virulence, colonization, infection	[21]
CckA	Function- specific effector	*C.crescentus*	---	4.7 μM	Affect cell cycle progression and cell replication	[125,126,127,128]
GlgX	Function- specific effector	*Streptomyces venezuelae*	C-terminus	~8 μM	Activate the catalytic activity of the enzyme to hydrolyze glycogen	[143]
MshE	Function- specific effector	*V. cholerae*	N-terminus	0.014–2 μM	Participate in MSHA pili assembly	[22,23,24]
HxrA	Function- specific effector	*P. aeruginosa*	N-terminus	---	Participate in MSHA pili assembly	[22,23,24]
LonA	Function- specific effector	*V. cholerae*	---	---	Regulate cell division, biofilm formation, flagellar motility	[144]
TfoY	Function- specific effector	*V. cholerae*	N-terminus	---	Regulate bacterial motility, participate in T6SS	[144]

The data presented in this table are compiled from various reported c-di-GMP effectors, which are categorized into five groups: RNA riboswitches, transcription factors, PilZ domain-containing proteins, GGDEF/EAL domain-containing proteins, and function-specific proteins. We have meticulously summarized the structural and functional characteristics of these effectors based on the latest available literature.

## Figures and Tables

**Figure 1 biomolecules-15-01471-f001:**
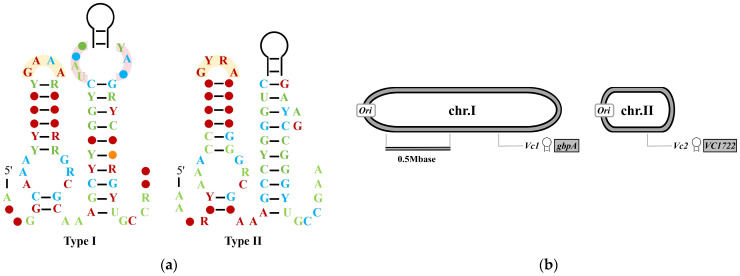
The Structure and Genomic Location of Type I Riboswitches. (**a**) Consensus sequences and structures of Type I and Type II GEMM RNAs. R (purine) represents adenine (A) or guanine (G); Y (pyrimidine) represents cytosine (C) or uracil (U). Different colors indicate the occurrence frequency of consensus bases in more than 500 type I riboswitch sequences: red (97%), blue (90%), green (75%), and orange (50%). The yellow arc area represents the GNRA or GYRA tetraloop, and the pink arc area represents the tetraloop receptor; (**b**) The Genomic locations of GEMM RNA sequences *Vc1* and *Vc2* in *Vibrio cholerae*. The two black-and-gray circles represent the two double-stranded circular chromosomes (chr.I and chr.II) of the *V. cholerae* genome, respectively. Adapted with permission from Ref. [37], 2017, ACS Publications and [38], 2025, Kavita, K.

**Figure 2 biomolecules-15-01471-f002:**
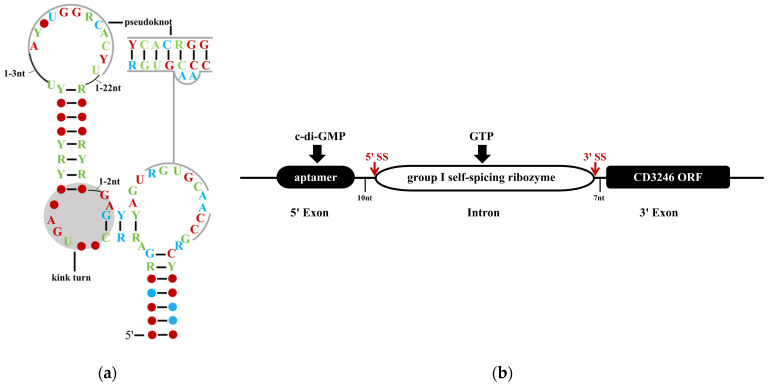
The Structure and Genomic Location of Type II Riboswitches. (**a**) Consensus sequences and structures of c-di-GMP-II. R (purine) represents adenine (A) or guanine (G); Y (pyrimidine) represents cytosine (C) or uracil (U). Different colors indicate the occurrence frequency of conserved bases in multiple representative type II riboswitch sequences: red (97%), blue (90%), and green (75%). The gray region represents the RNA kink turn, which is a conserved three-dimensional structural motif; (**b**) The Genomic Localization of Type II Riboswitches and Group I Self-Splicing Ribozymes in *Clostridioides difficile*. The red arrow denotes the splicing site. Adapted with permission from Ref. [37], 2017, ACS Publications and [38], 2025, Kavita, K.

**Figure 3 biomolecules-15-01471-f003:**
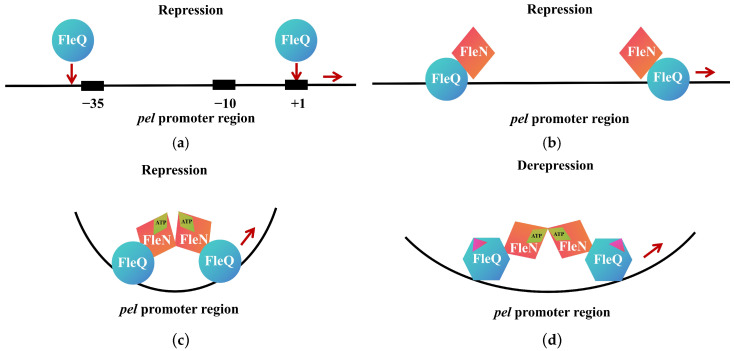
Schematic Diagram of FleQ Involved in the Regulation of *pel* Expression. (**a**) In the absence of c-di-GMP, FleQ binds to the FleQ-binding site on the *pel* promoter; (**b**) The interaction between FleQ and FleN in the absence of ATP; (**c**) In the presence of ATP, the interaction between FleQ and FleN causes structural distortion of *pel*; (**d**) In the presence of c-di-GMP, the binding of c-di-GMP to FleQ induces a conformational change in FleQ. This change, potentially transduced by FleN, alleviates the structural distortion of pel, thereby initiating the expression of *pel*. The magenta triangles represent c-di-GMP. Adapted with permission from Ref. [42], 2017, Oxford University Press.

**Figure 4 biomolecules-15-01471-f004:**
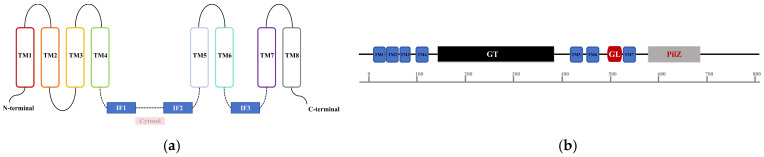
The Structure of BcsA from *Rhodobacter sphaeroides.* (**a**) Schematic diagram of the eight transmembrane helices in BcsA. TM represents transmembrane, IF represents interfacial helix, the cytoplasmic region is located between IF1 and IF2. Adapted with permission from Ref. [63], 2014, Elsevier Ltd.; (**b**) Secondary structure schematic for BcsA. The glycosyltransferase (GT) domain is located between TM4 and TM5, the “gating loop” (GL) between TM6 and TM7, and the PilZ domain within TM8.

**Figure 5 biomolecules-15-01471-f005:**
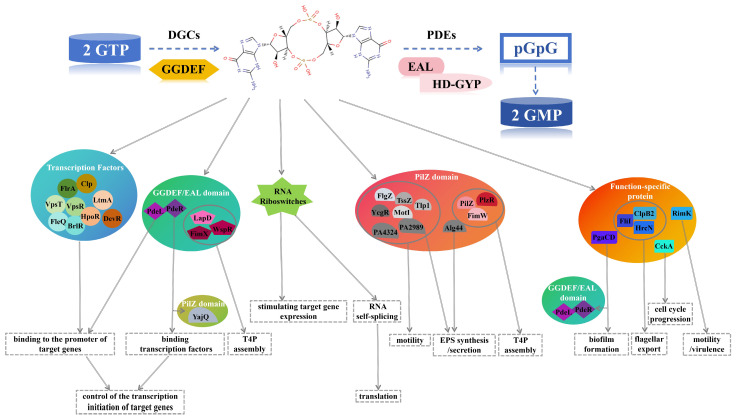
Diversity of c-di-GMP Effectors. Different types of effectors are represented by distinct patterns. Transcription factors FleQ, BrlR (from *Pseudomonas aeruginosa*), FlrA, VpsT, VpsR (from *Vibrio cholerae*), LtmA, HpoR, DevR (from *Mycobacterium smegmatis*), and the triggered phosphodiesterase PdeL act on the promoter regions of target genes. PdeR and the PilZ domain protein YajQ bind to transcription factors to regulate the transcriptional initiation of target genes. RNA riboswitches control target gene expression (Type I) and RNA self-splicing (Type II). PilZ domain proteins FlgZ, TssZ, YcgR, MotI, Tlp1, PA4324, and PA2989 affect bacterial motility; PA2989 and Alg44 participate in the regulation of EPS. PilZ, FimW, PlzR and GGDEF/EAL domain proteins (e.g., WspR, LapD, FimX) collectively regulate type IV pili synthesis. Triggered phosphodiesterases PdeL, PdeR and the functional specific protein PgaCD control biofilm formation. The output ATPases FliI, HrcN, ClpB2 regulate flagellar transport. The protein kinase/phosphatase CckA is involved in cell cycle progression, and the ribosome-modifying protein RimK influences bacterial motility and virulence.

## Data Availability

Not applicable.

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
