# Peer review of "Dynamic Executors of Bacterial Signals: Functional Versatility and Regulatory Networks of c-di-GMP Effectors"

_biomolecules, 2025, doi:10.3390/biom15101471_

Round 1

Reviewer 1 Report

Comments and Suggestions for Authors The manuscript by Jia et al. is devoted to the currently relevant topic of the functional diversity of c-di-GMP effectors in Gram-negative bacteria. As a result of active studies of c-di-GMP signaling and its regulation over the past fifteen years, a large amount of information has been accumulated, which is confirmed by regularly published reviews. I believe that readers would like to see in the introduction of the manuscript a clearer and more detailed justification of how, in the authors' opinion, their summary of the results obtained on this scientific topic fundamentally differs from those reflected, for example, in the reviews: Liu, C., Shi, R., Jensen, M. S., Zhu, J., Liu, J., Liu, X., ... & Liu, W. (2024). The global regulation of c‐di‐GMP and cAMP in bacteria. Mlife, 3(1), 42-56. Feng, Q., Zhou, J., Zhang, L., Fu, Y., & Yang, L. (2024). Insights into the molecular basis of c-di-GMP signaling in Pseudomonas aeruginosa. Critical Reviews in Microbiology, 50(1), 20-38. Homma, M., & Kojima, S. (2022). Roles of the second messenger c‐di‐GMP in bacteria: Focusing on the topics of flagellar regulation and Vibrio spp. Genes to Cells, 27(3), 157-172. I do not deny that the manuscript presents a critical analysis of a large amount of thematic literature, especially since it is focused on the prospect of using these results in the development of effective antibacterial drugs, for example, bifunctional inhibitors capable of simultaneously blocking the binding of effectors to both c-di-GMP and signaling molecules of the host organism. Moreover, most of the sources cited by the authors were published in the period 2004-2018. The authors note that the main goal of their study is to summarize the structural and functional diversity of c-di-GMP effectors to understand the adaptive regulatory mechanisms of bacteria, which may be in demand in the development of new antibacterial drugs. However, the abstract of the manuscript emphasizes that Pseudomonas aeruginosa, Vibrio cholerae and Escherichia coli will be considered as model bacteria. However, the article presents the results of studies covering a wider range of both Gram-negative and Gram-positive bacteria. Section 2.2 - Genus Clostridioides Lines 205–229 Mycobacterium Lines 229–236 Xanthomonas campestris Lines 244 Gluconacetobacter xylinum Lines 311–336 Salmonella enterica, Bacillus subtilis, C. crescentus, Lysobacter enzymogenes, Xanthomonas campestris Lines 169 and 273 of the Latin alphabet should be italicized. Table 1. Sometimes the authors make comparisons with the target bacteria, and then this is justified, but sometimes they simply state facts about other bacteria. I think the authors should edit the text, since most of the manuscript is devoted to describing the c-di-GMP signaling pathways in three target microorganisms. Or remove the emphasis that the data are considered on three model bacteria. I believe that for better perception of the generalized material on different types of bacterial c-di-GMP effectors in the table the order of data presentation should be slightly changed. The first column should be devoted to bacterial species and the data should be grouped by bacterial species, so that their evolutionary differences can be noted. Figure 3 If you arrange the effectors, preserving their patterns, around c-di-GMP, this will allow you to enlarge the labels somewhat and improve the perception of the data. What means “molity”? The conclusion about the need to combine dynamic visualization methods, computational biology and synthetic biology for systematic deciphering of multidimensional regulatory networks of effectors is quite general. Moreover, even identifying all the details of regulatory networks of effectors, taking into account their number, is unlikely to allow developing a universal scheme of effective antibacterial therapy. However, I like the style of the article, its logic, the structure of the material. I believe that the review can be published after editing and will be interesting to readers.

Author Response

Comments 1:

The manuscript by Jia et al. is devoted to the currently relevant topic of the functional diversity of c-di-GMP effectors in Gram-negative bacteria. As a result of active studies of c-di-GMP signaling and its regulation over the past fifteen years, a large amount of information has been accumulated, which is confirmed by regularly published reviews. I believe that readers would like to see in the introduction of the manuscript a clearer and more detailed justification of how, in the authors' opinion, their summary of the results obtained on this scientific topic fundamentally differs from those reflected, for example, in the reviews: Liu, C., Shi, R., Jensen, M. S., Zhu, J., Liu, J., Liu, X., ... & Liu, W. (2024). The global regulation of c‐di‐GMP and cAMP in bacteria. Mlife, 3(1), 42-56. Feng, Q., Zhou, J., Zhang, L., Fu, Y., & Yang, L. (2024). Insights into the molecular basis of c-di-GMP signaling in Pseudomonas aeruginosa. Critical Reviews in Microbiology, 50(1), 20-38. Homma, M., & Kojima, S. (2022). Roles of the second messenger c‐di‐GMP in bacteria: Focusing on the topics of flagellar regulation and Vibrio spp. Genes to Cells, 27(3), 157-172.

Response 1:

Thank you for pointing this out. We agree with this comment. Therefore, we have made the following comparisons between the three papers you mentioned and our review from the perspective of their core themes.

  1. The article by Liu et al. primarily elaborates on the global regulatory models of two signaling molecules, cyclic adenosine monophosphate (cAMP) and cyclic diguanylate monophosphate (c-di-GMP). It compares their distinct regulatory frameworks and explores the mechanisms and physiological significance of cross-regulation between c-di-GMP and cAMP. Specifically, it focuses on the mechanisms of action and physiological functions of the following components: transcription factors including FleQ (from Pseudomonas aeruginosa), VpsT and Vps (from Vibrio cholerae), and RsiG (from Streptomyces venezuelae); phosphodiesterase LapD (from Pseudomonas aeruginosa); RpfR (from Burkholderia cepacia); function-specific kinases ShkA and CccK (from Caulobacter crescentus); translation elongation factors (from Acinetobacter baumannii); and deacetylase CobB (from Escherichia coli).
  2. The article by Feng et al. explores the potential mechanisms underlying the cyclic diguanylate monophosphate (c-di-GMP) signaling network in Pseudomonas aeruginosa. It primarily focuses on four aspects: how environmental cues regulate c-di-GMP signaling, function regulation mediated by protein-protein interactions, the heterogeneity of c-di-GMP, and the crosstalk between c-di-GMP and other signaling systems.
  3. The article by Homma et al. focuses on investigating the role of cyclic diguanylate monophosphate (c-di-GMP) in the genus Vibrio(with an emphasis on Vibrio cholerae) and flagellum-related fields.

We acknowledge that some viewpoints in the aforementioned articles overlap with those of this review, while we also note that certain effectors among them have not been included in the discussion scope of this review. This review classifies effectors into a total of five categories based on their structure or function. Taking these categories as the foundation, it conducts a systematic summary of representative effectors from different bacterial species under each category, elaborates on their key roles in bacterial pathogenicity, antibiotic tolerance, and host-pathogen interactions, and thereby provides innovative research insights for in-depth studies on the regulatory mechanisms of c-di-GMP signaling. Additionally, it is worth noting that the effectors covered in this review only a small portion of the relevant research, which does not diminish the importance of those unmentioned effectors.

Based on the above explanations, we have finally made the following revisions: “Therefore, in this review, starting from the currently reported types of c-di-GMP effectors, we classify them into five categories based on differences in their structure or function. We then systematically and comprehensively summarizes the structural and functional diversity of these effectors across various bacterial species, thereby providing theoretical support for the analysis of bacterial adaptive regulatory mechanisms and the development of novel antimicrobial therapies.” Page 2, Introduction, Line 79-84.

Comments 2:

I do not deny that the manuscript presents a critical analysis of a large amount of thematic literature, especially since it is focused on the prospect of using these results in the development of effective antibacterial drugs, for example, bifunctional inhibitors capable of simultaneously blocking the binding of effectors to both c-di-GMP and signaling molecules of the host organism. Moreover, most of the sources cited by the authors were published in the period 2004-2018.

Response 2:

Thank you for pointing this out. We agree with this comment. Therefore, based on the needs of the manuscript, we have appropriately added several references published between 2019 and 2025. The modified references are “#81, #82, #142, and #143.” Page 24, References, Line 804-809 and Page 24, References, Line 960-964.

Comments 3:

The authors note that the main goal of their study is to summarize the structural and functional diversity of c-di-GMP effectors to understand the adaptive regulatory mechanisms of bacteria, which may be in demand in the development of new antibacterial drugs. However, the abstract of the manuscript emphasizes that Pseudomonas aeruginosa, Vibrio cholerae and Escherichia coli will be considered as model bacteria. However, the article presents the results of studies covering a wider range of both Gram-negative and Gram-positive bacteria. Section 2.2 - Genus Clostridioides Lines 205–229 Mycobacterium Lines 229–236 Xanthomonas campestris Lines 244 Gluconacetobacter xylinum Lines 311–336 Salmonella enterica, Bacillus subtilis, C. crescentus, Lysobacter enzymogenes, Xanthomonas campestris.

Response 3:

Thank you for pointing this out. We agree with this comment. Therefore, we have replaced the expression “using Pseudomonas aeruginosa, Vibrio cholerae, Escherichia coli as model systems” with “using several representative bacterial species as model systems”. Page 1, Abstract, Line 23.

Comments 4:

Lines 169 and 273 of the Latin alphabet should be italicized.

Response 4:

Thank you for pointing this out. We agree with this comment. We have revised it: “FleN (PA1454), another regulator of flagellar genes in P. aeruginosa. Page 6, C-di-GMP-Responsive Transcription Factors: Structural Determinants and Regulatory Roles in Bacterial Physiology, Line 184.

Comments 5:

Sometimes the authors make comparisons with the target bacteria, and then this is justified, but sometimes they simply state facts about other bacteria. I think the authors should edit the text, since most of the manuscript is devoted to describing the c-di-GMP signaling pathways in three target microorganisms. Or remove the emphasis that the data are considered on three model bacteria.

Response 5:

Thank you for pointing this out. We agree with this comment. This review discusses representative effectors across a variety of bacterial species. Therefore, We have deleted the expression “using Pseudomonas aeruginosa, Vibrio cholerae, Escherichia coli as model systems” in accordance with your suggestions.

Comments 6:

Table 1. I believe that for better perception of the generalized material on different types of bacterial c-di-GMP effectors in the table the order of data presentation should be slightly changed. The first column should be devoted to bacterial species and the data should be grouped by bacterial species, so that their evolutionary differences can be noted. 

Response 6:

Thank you for pointing this out. While your suggestions are constructive, this review takes different types of effectors as its core starting point. Therefore, we have placed the names and types of effectors in the first two columns of the table, and this arrangement is consistent with our core logic. Thus, we kindly request your permission to retain the original layout of the Table 1.

Comments 7:

Figure 3. If you arrange the effectors, preserving their patterns, around c-di-GMP, this will allow you to enlarge the labels somewhat and improve the perception of the data. What means “molity”?

Response 7:

Thank you for pointing this out. We agree with this comment and sincerely apologize for the spelling errors. We have revised Figure 3 (Figure 5 in revised manuscript) in accordance with your suggestions in revised manuscript.

Figure 5

Comments 8:

The conclusion about the need to combine dynamic visualization methods, computational biology and synthetic biology for systematic deciphering of multidimensional regulatory networks of effectors is quite general. Moreover, even identifying all the details of regulatory networks of effectors, taking into account their number, is unlikely to allow developing a universal scheme of effective antibacterial therapy. 

Response 8:

Thank you for pointing this out. We agree with this comment. Therefore, we have provided a detailed discussion on the potential future research directions and methodologies for effector molecules, as well as therapeutic approaches for bacterial infections related to these effectors: “In future studies, live-cell dynamic imaging technology can be specifically integrated to achieve full-process tracking of the spatiotemporal localization and dynamic interactions of effectors within bacterial hosts to the greatest extent possible. Computational biology approaches should be employed to construct interaction models between effectors and host target proteins, thereby deciphering the regulatory mechanisms mediated by allosteric effects. Additionally, synthetic biology tools can be utilized to validate the functions of effectors by reconstructing or interfering with their action pathways. Based on the combined application of the aforementioned technologies, it will be feasible to systematically decode the multi-dimensional regulatory networks of effectors at the levels of “protein interaction—signal pathway regulation—bacterial pathogenic phenotype”. Ultimately, this will provide theoretical and technical support for the development of new paradigms for the precise intervention of bacterial infections, such as “small-molecule inhibitors targeting key interaction sites of effectors” and “gene therapy strategies blocking effector regulatory pathways””. Page 15, Conclusions and Outlook, Line 575-589.

Reviewer 2 Report

Comments and Suggestions for Authors

The manuscript aims to review current data on сyclic di-GMP, an important secondary messenger in the life of bacteria. The review lacks logical connections and generalizations and is therefore difficult to read. Figures 1 and 2 contain inaccuracies and need revision as indicated below:

Lines 32-33: According to modern classification of bacteria in NCBI and updated Genome Taxonomy Database (GTDB)  it is more correctly to indicate the cited bacterial strain as Gluconacetobacter xylinus (modern synonym Komagataeibacter xylinus)

Lines 109-113: It seems as the colors of arc area are not in agree with that described in the figure-1 caption. The color claimed as orange for indication riboswitch sequences looks like yellow. It is better to change these colors.

Signs in Figures 1 and 2 (section (b)) are better to enlarge.

Lines 144-150: It is not clear what does the gray colour shows in Figure 2. This should be explained.

Line 196: "can binds" should be corrected as "can bind"

Also, the author is advised to prepare more popular illustrations for each subsection to make the content easier to perceive through visualization. In addition, it would be very useful to support a complex text composed of listed facts with more detailed, even popular, descriptive paragraphs to make the content more complete and understandable. In general, the conclusions in this article are acceptable.

Author Response

Comments 1:

Lines 32-33: According to modern classification of bacteria in NCBI and updated Genome Taxonomy Database (GTDB) it is more correctly to indicate the cited bacterial strain as Gluconacetobacter xylinus (modern synonym Komagataeibacter xylinus).

Response 1:

Thank you for pointing this out. We agree with this comment. we have replaced the expression “Gluconacetobacter xylinus” with “Komagataeibacter xylinus” or “K. xylinus”. Page 1, Introduction, Line 33-34. Page 7, The PilZ Domain Fold: C-di-GMP Binding and Allosteric Regulation, Line 262. Page 1, Introduction, Line 33-34. Page 17, Table 1.

Comments 2:

Lines 109-113: It seems as the colors of arc area are not in agree with that described in the figure-1 caption. The color claimed as orange for indication riboswitch sequences looks like yellow. It is better to change these colors.

Response 2:

Thank you for pointing this out. We agree with this comment. We have adjusted the color in Figure 1 as per your suggestions.

Figure 1(a)

Comments 3:

Signs in Figures 1 and 2 (section (b)) are better to enlarge.

Response 3:

Thank you for pointing this out. We agree with this comment. We have adjusted the signs in Figure 1 and 2(section (b)) as per your suggestions.

Figure 1(b)

Figure 2(b)

Comments 4:

Lines 144-150: It is not clear what does the gray colour shows in Figure 2. This should be explained.

Response 4:

Thank you for pointing this out. We agree with this comment. Therefore, We have revised the captions for Figure 1 and Figure 2 respectively to make the information in the figures more comprehensive. Figure 1(b) The Genomic locations of GEMM RNA sequences Vc1 and Vc2 in Vibrio cholerae. The two black-and-gray circles represent the two double-stranded circular chromosomes (chr.I and chr.II) of the V. cholerae genome, respectively. Page 4, 2.1 C-di-GMP-I Riboswitches: Structural Basis for Regulating Pathogenic Gene Expression in V. cholerae, Line 127-128. Figure 2(a) The gray region represents the RNA kink turn, which is a conserved three-dimensional structural motif. Page 5, 2.2 C-di-GMP-II Riboswitches: Controlling GTP-Mediated RNA Self-Splicing in Clostridioides difficile, Line 165-166.

Comments 5:

Line 196: "can binds" should be corrected as "can bind".

Response 5:

Thank you for pointing this out. We agree with this comment. We apologize for this grammatical error and have made the necessary corrections. “VpsR, another NtrC-family transcription factor akin to FleQ, can bind c-di-GMP with a KD of 1.6 μM.” Page 6, 3. C-di-GMP-Responsive Transcription Factors: Structural Determinants and Regulatory Roles in Bacterial Physiology, Line 215.

Comments 6:

Also, the author is advised to prepare more popular illustrations for each subsection to make the content easier to perceive through visualization. In addition, it would be very useful to support a complex text composed of listed facts with more detailed, even popular, descriptive paragraphs to make the content more complete and understandable.

Response 6:

Thank you for pointing this out. We agree with this comment. Therefore, we have appropriately added some schematic diagrams and included additional detailed descriptive paragraphs, which have been marked in the revised manuscript.

Reviewer 3 Report

Comments and Suggestions for Authors

The text is a very comprehensive review and of great interest to the community. My observations focus on specific nomenclature clarifications, methodological clarifications, and organizational suggestions. I recommend including some figures or diagrams to facilitate understanding of the more technical sections. I consider this work to be solid and can be accepted after making the necessary minor corrections.

Introduction
Lines 32–33: Please update the name of Gluconacetobacter xylinum to Komagataeibacter xylinus.
Lines 70–71: "...c-di-GMP exhibits immunomodulatory effects in animals or humans..." If this is done later, I will omit this comment, but I think it would improve the quality of the text if you enriched it by describing the models and results.
RNA-Based c-di-GMP Effectors: Riboswitches and Their Mechanisms

2.1: C-di-GMP-I Riboswitches
Lines 91–92: “...GEMM (Genes for the Environment, Membranes and Motility)...” I suggest clarifying that this is a family within the type I riboswitches.
Line 98–100: “Vc2 forms a binary saturated complex with c-di-GMP (KD = 1 nM)...” Please include the method used.
Lines 89–115: The section is very complete and well-developed. However, it would benefit from expanding the discussion of its in vivo physiological impact.

2.2 C-di-GMP-II Riboswitches
Lines 125–130: I find this research very interesting and appropriate to include in the text. However, I think it is important to explain what such a high affinity means functionally.

3. C-di-GMP-Responsive Transcription Factors
Lines 169–180: Please clarify whether FleN has its own affinity for c-di-GMP or if its function is exclusively cooperative.
Lines 153–236: Although not mandatory, I believe that the addition of a figure depicting how these factors relate to the modulation of different bacterial functions would greatly improve the quality of the text. .

4. The PilZ Domain Fold: C-di-GMP Binding and Allosteric Regulation
Lines 248–252: I consider it important to add an image to facilitate reading comprehension, as these are highly specialized topics.
Lines 270–271: Please clarify whether the validation was done through functional assays or only in vitro.
Lines 272–305: This section lists too many examples (PilZ, FlgZ, MapZ, HapZ, PA4324, TssZ, FimW, PA2989) and may be difficult for the reader to understand. I think it would be more appropriate to separate them into subtopics.

6: The Role of c-di-GMP Effectors with High Functional Specificity in Bacterial Life Activities
Lines 424–426: “...leads to irreversible inactivation of the Pga system...” Please explain why this is physiologically relevant.
Line 443–444: “c-di-GMP likely binds to a pocket of highly conserved residues at the interface of the two FliI subunits...” Although already mentioned, please explicitly clarify that this is an in silico prediction, not direct structuring.

Author Response

Comments 1:

Lines 32-33: Please update the name of Gluconacetobacter xylinum to Komagataeibacter xylinus.

Response 1:

Thank you for pointing this out. We agree with this comment. we have replaced the expression “Gluconacetobacter xylinus” with “Komagataeibacter xylinus” or “K. xylinus”. Page 1, Introduction, Line 33-34. Page 7, The PilZ Domain Fold: C-di-GMP Binding and Allosteric Regulation, Line 262. Page 1, Introduction, Line 33-34. Page 17, Table 1.

Comments 2:

Lines 70-71: "...c-di-GMP exhibits immunomodulatory effects in animals or humans..." If this is done later, I will omit this comment, but I think it would improve the quality of the text if you enriched it by describing the models and results.

Response 2:

Thank you for pointing this out. We agree with this comment. Therefore, We have supplemented the text with a brief description of the interaction mechanism between the effector STING and c-di-GMP in mammals, providing a basis for the theory proposed earlier. “For instance, stimulator of interferon genes (STING) can induce a type I interferon(IFN) response by binding to c-di-GMP, thereby functioning as an immune sensor. Moreover, the binding of STING to c-di-GMP  can enhance the interaction between the C-terminal domain of STING and TANK-binding kinase 1 (TBK1), ultimately promoting the production of type I interferons.” Page 2, Introduction, Line 74-79.

Comments 3:

Lines 91-92: “...GEMM (Genes for the Environment, Membranes and Motility)...” I suggest clarifying that this is a family within the type I riboswitches.

Response 3:

Thank you for pointing this out. We agree with this comment. Therefore, We have provided a clear explanation of GEMM in accordance with your suggestions. “The high conservation and genomic distribution characteristics exhibited by GEMM RNA are typical features of c-di-GMP-I riboswitches.” Page 3, 2.1 C-di-GMP-I Riboswitches: Structural Basis for Regulating Pathogenic Gene Expression in V. cholerae, Line 101-102.

Comments 4:

Line 98-100: “Vc2 forms a binary saturated complex with c-di-GMP (KD = 1 nM)...” Please include the method used.

Response 4:

Thank you for pointing this out. We agree with this comment. Therefore, We have added the determination methods for the two KD values respectively. “Vc2 forms a binary saturated complex with c-di-GMP (KD = 1 nM, determined by Fluorescence Titration Method) exhibiting tighter binding affinity than the E. coli PilZ domain (KD = 840 nM, determined by Equilibrium Dialysis). “ Page 3, 2.1 C-di-GMP-I Riboswitches: Structural Basis for Regulating Pathogenic Gene Expression in V. cholerae, Line 108-111.

Comments 5:

Lines 89-115: The section is very complete and well-developed. However, it would benefit from expanding the discussion of its in vivo physiological impact.

Response 5:

Thank you for pointing this out. We agree with this comment. Therefore, we have provided a more detailed explanation of the physiological functions of this type of riboswitch. Pathogenic V. cholerae harbors two Type I GEMM RNA sequences in its genome: “one (Vc1) is located upstream of the gbpA gene, and the other (Vc2) is situated upstream of VC1722, a homologous gene of tfoX (a gene encoding a global transcription factor)(Figure 1b). “ “When the intracellular concentration of c-di-GMP in bacteria increases, the Vc2 riboswitch triggers higher expression of pathogenicity-related genes. Therefore, Vc2 is referred to as a “gene-on switch”.” “the Vc1 riboswitch senses VieA-mediated c-di-GMP depletion to promote the virulence-associated gene gbpA expression and initiate infection.” Page 3, 2.1 C-di-GMP-I Riboswitches: Structural Basis for Regulating Pathogenic Gene Expression in V. cholerae, Line 105-108, 111-113, 119.

Comments 6:

Lines 125–130: I find this research very interesting and appropriate to include in the text. However, I think it is important to explain what such a high affinity means functionally.

Response 6:

Thank you for pointing this out. We agree with this comment. Therefore, We have described the physiological function of the high affinity between c-di-GMP and c-di-GMP-II. “We propose that this high affinity can minimize the probability of “regulatory errors”, ensuring that c-di-GMP signals are stably transmitted to downstream genes, thereby safeguarding the adaptability or pathogenicity of bacteria in complex environments.” Page 4, 2.2 C-di-GMP-II Riboswitches: Controlling GTP-Mediated RNA Self-Splicing in Clostridioides difficile, Line 142-145.

Comments 7:

Lines 169-180: Please clarify whether FleN has its own affinity for c-di-GMP or if its function is exclusively cooperative.

Response 7:

Thank you for pointing this out. We agree with this comment. Therefore, We have provided additional explanations regarding the affinity between FleN and c-di-GMP. “FleN (PA1454), another regulator of flagellar genes in P. aeruginosa, is an intracellular ATPase containing an atypical Walker A motif and lacks direct c-di-GMP binding activity.” Page 6, 3. C-di-GMP-Responsive Transcription Factors: Structural Determinants and Regulatory Roles in Bacterial Physiology, Line 188-189.

Comments 8:

Lines 153-236: Although not mandatory, I believe that the addition of a figure depicting how these factors relate to the modulation of different bacterial functions would greatly improve the quality of the text.

Response 8:

Thank you for pointing this out. We agree with this comment. Therefore, We have included a schematic diagram of the bacterial function regulatory mechanism by FleQ as a representative, while the physiological functions of other transcription factors are presented in Table 1.

Comments 9:

Lines 248-252: I consider it important to add an image to facilitate reading comprehension, as these are highly specialized topics.

Response 9:

Thank you for pointing this out. We agree with this comment. Therefore, we have supplemented the structural schematic diagram of BcsA.

Comments 10:

Lines 270-271: Please clarify whether the validation was done through functional assays or only in vitro.

Response 10:

Thank you for pointing this out. We agree with this comment. Therefore, We have supplemented the method for the conserved amino acid mutation experiment of Alg44. “After Merighi et al. mutated several conserved amino acids (R17, R21, D44, S46) in its N-terminal PilZ domain to alanine, they used the filter binding assay to determine the binding ability of the Alg44mutant (Alg44R17A, Alg44R17A,R21A, Alg44D44A, Alg44R21A) to c-di-GMP in vitro, and also detected the alginate phenotype of these mutants. It was found that the Alg44mutant lost their c-di-GMP binding ability and failed to produce alginate.” Page 8, 4. The PilZ Domain Fold: C-di-GMP Binding and Allosteric Regulation, Line 288-294.

Comments 11:

Lines 272-305: This section lists too many examples (PilZ, FlgZ, MapZ, HapZ, PA4324, TssZ, FimW, PA2989) and may be difficult for the reader to understand. I think it would be more appropriate to separate them into subtopics.

Response 11:

Thank you for pointing this out. We agree with this comment. However, there is some overlap and integration in the content when we describe these effectors, making it difficult to split them completely into several subtopics. Furthermore, we have already classified these proteins based on their structural and functional characteristics in Figure 3 (Figure 5 in revised manuscript) and Table 1; therefore, we kindly request that the original description format be retained.

Comments 12:

Lines 424-426: “...leads to irreversible inactivation of the Pga system...” Please explain why this is physiologically relevant.

Response 12:

Thank you for pointing this out. We agree with this comment. Therefore, we have supplemented the explanation of the impact of irreversible inactivation of the Pga system on bacterial physiological functions. “At low c-di-GMP concentrations, PgaD exhibits instability, fails to bind to PgaC and is rapidly degraded. Consequently, when cellular c-di-GMP levels decline, the rapid turnover of PgaD leads to irreversible inactivation of the Pga system (which is responsible for the synthesis and secretion of poly-GlcNAc), temporarily decoupling the synthesis and secretion of poly-GlcNAc from the regulation of c-di-GMP levels. At this point, the cell needs to activate the derepressed carbon storage regulator (Csr) pathway   to support the resynthesis of all Pga components, thereby restoring the synthesis of poly-N-acetylglucosamine (poly-GlcNAc). This mechanism ensures that bacteria can still maintain normal life activities when short-term fluctuations occur in c-di-GMP levels.” 

Comments 13:

Line 443–444: “c-di-GMP likely binds to a pocket of highly conserved residues at the interface of the two FliI subunits...” Although already mentioned, please explicitly clarify that this is an in silico prediction, not direct structuring.

Response 12:

Thank you for pointing this out. We agree with this comment. Therefore, to avoid redundant expression, we have only replaced the periods with colons to provide explanations for the results. “Subsequently, the binding sites between the two were further predicted using a combination of mass spectrometric analysis and in silico simulations: c-di-GMP likely binds to a pocket of highly conserved residues at the interface of the two FliI subunits.”

Round 2

Reviewer 2 Report

Comments and Suggestions for Authors

Accept in present form